# Effect of immune modulation on the skeletal muscle mitochondrial exercise response: An exploratory study in mice with cancer

**Linda A. Buss**[1¤a]*, **Barry Hock**[2], **Troy L. Merry**[3,4], **Abel D. Ang**[1¤b], **Bridget A. Robinson**[1,5], **Margaret J. Currie**[1], **Gabi U. Dachs**[1]

**1** Mackenzie Cancer Research Group, Department of Pathology and Biomedical Science, University of Otago, Christchurch, New Zealand, **2** Hematology Research Group, Department of Pathology and Biomedical Science, University of Otago, Christchurch, New Zealand, **3** Discipline of Nutrition, Faculty of Medical and Health Sciences, University of Auckland, Auckland, New Zealand, **4** Maurice Wilkins Centre for Molecular Biodiscovery, University of Auckland, Auckland, New Zealand, **5** Canterbury Regional Cancer and Hematology Service, Canterbury District Health Board, Christchurch, New Zealand

¤a Current address: Christchurch Heart Institute, Department of Medicine, University of Otago, Christchurch, New Zealand
¤b Current address: School of Medicine, University of Dundee, Dundee, United Kingdom
* linda.buss@otago.ac.nz

**Data Availability Statement:** All relevant data are within the manuscript and its Supporting Information files.

## Abstract

Cancer causes mitochondrial alterations in skeletal muscle, which may progress to muscle wasting and, ultimately, to cancer cachexia. Understanding how exercise adaptations are altered by cancer and cancer treatment is important for the effective design of exercise interventions aimed at improving cancer outcomes. We conducted an exploratory study to investigate how tumor burden and cancer immunotherapy treatment (anti-PD-1) modify the skeletal muscle mitochondrial response to exercise training in mice with transplantable tumors (B16-F10 melanoma and EO771 breast cancer). Mice remained sedentary or were provided with running wheels for ~19 days immediately following tumor implant while receiving no treatment (Untreated), isotype control antibody (IgG2a) or anti-PD-1. Exercise and anti-PD-1 did not alter the growth rate of either tumor type, either alone or in combination therapy. Untreated mice with B16-F10 tumors showed increases in most measured markers of skeletal muscle mitochondrial content following exercise training, as did anti-PD-1-treated mice, suggesting increased mitochondrial content following exercise training in these groups. However, mice with B16-F10 tumors receiving the isotype control antibody did not exhibit increased skeletal muscle mitochondrial content following exercise. In untreated mice with EO771 tumors, only citrate synthase activity and complex IV activity were increased following exercise. In contrast, IgG2a and anti-PD-1-treated groups both showed robust increases in most measured markers following exercise. These results indicate that in mice with B16-F10 tumors, IgG2a administration prevents exercise adaptation of skeletal muscle mitochondria, but adaptation remains intact in mice receiving anti-PD-1. In mice with EO771 tumors, both IgG2a and anti-PD-1-treated mice show robust skeletal muscle mitochondrial exercise responses, while untreated mice do not. Taken together, we postulate that immune modulation may enhance skeletal muscle mitochondrial response to exercise

**Funding:** We appreciate funding from the Mackenzie Charitable Foundation (MJC, GUD, BAR), the University of Otago (doctoral scholarship for LAB) and the McGee Fellowship Fund (LAB). The funders had no role in study design, data collection and analysis, decision to publish, or preparation of the manuscript.

**Competing interests:** The authors have declared that no competing interests exist.

**Abbreviations:** COX-IV, Cytochrome c oxidase subunit IV; PD-1, programmed death protein 1; OXPHOS, oxidative phosphorylation; CI-V, complexes I-V; cyt c, cytochrome c; GAPDH, glyceraldehyde 3-phosphate dehydrogenase; DTT, dithiothreitol; PBS/TBS, phosphate/tris-buffered saline; ANOVA, analysis of variance; ANCOVA, analysis of covariance.

in tumor-bearing mice, and suggest this as an exciting new avenue for future research in exercise oncology.

## Introduction

Cancer induces systemic metabolic perturbations that often lead to declines in muscle mass and physical function [1]. Mitochondrial impairments are seen in skeletal muscle prior to any noticeable muscle wasting in both mice [2] and humans [3], suggesting that mitochondrial dysfunction may play a role in the pathogenesis of cancer cachexia and serve as an early indicator of its onset. In contrast, exercise induces improvements in muscle mass and function, including increased mitochondrial content [4]. This is reflected by increased protein levels and activity of the oxidative phosphorylation (OXPHOS) complexes, and increased activity of metabolic enzymes such as citrate synthase [5]. As such, exercise may be able to attenuate cancer-induced impairments in muscle (mitochondrial) function and the development of cancer cachexia [6, 7]. In addition, exercise has been shown to improve anxious and depressive symptoms in cancer patients, lower fatigue levels, reduce treatment-related side effects and has been associated with improved survival [8–10]. As a result, guidelines are being developed for the implementation of exercise into oncology care [11, 12].

However, it is currently unknown whether cancer modifies exercise-induced skeletal muscle mitochondrial adaptations. Given that mitochondrial impairments are seen in skeletal muscle prior to any noticeable muscle wasting [2, 3], it would be unwise to assume that exercise induces muscular adaptations with the same efficiency in people with cancer as in healthy individuals. Understanding how tumor burden and cancer treatment affect exercise adaptations is important to inform the development of exercise programs for cancer patients.

Muscle regeneration and repair following damage, including ultrastructural damage caused by exercise [13], is highly dependent on a tightly controlled sequence of events orchestrated by the immune system [14, 15]. Initially, localized inflammation occurs, followed by an anti-inflammatory shift at 1–3 days post-injury to facilitate tissue repair [14]. Although current knowledge is largely extrapolated from more severe models of muscle damage such as cardiotoxin injection [13], histological evidence of leukocyte accumulation in human muscle following exercise has been reported [16]. In addition, exercise induced the gene expression of macrophage-associated chemokines and cytokines in muscle following exercise [17], and anti-inflammatory medication may modulate muscle adaptation to exercise [18]. This suggests that local immune responses in skeletal muscle may be important for muscle adaptation to exercise training.

Tumor burden induces local and systemic immune dysfunction, impeding effective anti-tumor immunity [19]. Immune checkpoint inhibitors (monoclonal antibodies against targets such as programmed death protein 1, PD-1) are a relatively new class of immunotherapy drug increasingly used in the treatment of cancers such as advanced melanoma. They are aimed at reactivation of the anti-tumor immune response, leading to sustained, long-term remissions in some patients [20]. Due to the role of the immune system in muscle regeneration and adaptation to exercise (as discussed in the previous paragraph) [13, 14], it is possible that immunotherapies such as anti-PD-1 will affect muscle responsiveness to exercise training. Given the recent explosion of immunotherapy in the treatment of cancer and accumulating evidence for the benefits of exercise in oncology care, it is important to understand the interactions between the tumor, immunotherapy and exercise.

Due to the lack of existing data regarding the effect of tumor burden on the immune landscape in skeletal muscle, we did not have a preconceived hypothesis regarding the effect of immunotherapy on mitochondrial exercise adaptations in muscle. Therefore, we conducted an exploratory analysis on existing cohorts of mice, investigating how tumor burden and anti-PD-1 treatment impact skeletal muscle mitochondrial response to exercise by measuring changes in mitochondrial markers following exercise training of tumor-free and tumor-bearing (B16-F10 melanoma and EO771 breast cancer) mice receiving anti-PD-1 therapy.

## Material and methods

### Mouse model and ethics

Ethical approval was obtained from the University of Otago Animal Ethics Committee (C01/16, C04/17, AUP-18-144, AUP-18-179). International guidelines on animal welfare in experimental neoplasia were strictly followed [21]. Female C57BL/6 mice aged 6–10 weeks were used for all experiments. This age range is frequently used in murine oncology studies [22–24]. Mice were housed in pairs with a cage divider in a standard rat cage (floor area 904 cm$^2$) under a 12:12 hour light:dark cycle. The temperature was maintained around 22°C. Mice were kept on a normal chow diet, provided *ad libitum* along with water. Exercising mice were provided with a modified Fast-Trac$^{TM}$ saucer wheel (Bio-Serv, Flemington, NJ, USA). Running wheels were equipped with a magnetic sensor and digital counter to quantify revolutions. These were designed and purpose-built by Mr Andrew Dachs (Decision Consulting Ltd, NZ; https://github.com/wirebadger/mouse-wheel).

### Tumor models

B16-F10 (American Type Culture Collection, ATCC) and EO771 (gifted by A/Prof Andreas Moeller, QIMR Berghofer, Australia) cells were cultured in Dulbecco's Modified Eagle Medium (DMEM) with GlutaMax and L-D glucose (Gibco Invitrogen, Carlsbad, CA, USA), supplemented with 10% fetal calf serum (FCS, Gibco). Cells used for implant had a maximum passage number of 19. Mycoplasma testing is routinely performed in our laboratory.

At experiment begin, 6-10-week-old tumor-bearing mice were inoculated with either $10^6$ B16-F10 melanoma cells in 50 μL PBS, delivered subcutaneously into the right flank or $2x10^5$ EO771 breast cancer cells in 20 μL phosphate buffered saline (PBS, 0.137 M NaCl, 2.7 mM KCl, 10.1 mM $Na_2HPO_4$, 1.8 mM $KH_2PO_4$), delivered into the 4$^{th}$ mammary fat pad.

### Study design

Experiments were performed on three cohorts of mice and conducted at different time-points which are outlined below. This means that we cannot exclude baseline differences in mitochondrial marker expression and therefore have not statistically compared results across these cohorts.

**Tumor-free mice (cohort 1).** Tumor-free mice were randomized into exercise or no exercise at 6–10 weeks of age and euthanized at 19 days post-experiment begin. This time-point was selected as it was the median time to reach maximum tumor size in the tumor-bearing mice. Mice were anaesthetized by isoflurane (Baxter, Deerfield, IL, USA) inhalation and euthanized by cervical dislocation. The left quadriceps femoris muscle was removed and frozen at -80°C.

**Untreated, tumor-bearing mice (cohort 2).** At tumor implant, mice were randomized into exercise (voluntary wheel running) or no exercise (no wheel). When tumor volume reached the ethical limit of 600 mm$^3$ (EO771) or 1000 mm$^3$ (B16-F10) or the welfare of the

mouse was impacted (by tumor burden, ulceration of the tumor or suspicion of internal tumors in mice with EO771 tumors), mice were anaesthetized by isoflurane (Baxter, Deerfield, IL, USA) inhalation and euthanized by cervical dislocation. The left quadriceps femoris muscle was removed and frozen at -80˚C. Tumor growth and tumor characteristics for this cohort has previously been published [22].

**Checkpoint inhibitor-treated, tumor-bearing mice (cohort 3).** Tumor initiation and endpoint, exercise and euthanasia were conducted as above. Additionally, when tumors reached 50–100 $mm^3$, twice-weekly treatment with 200 μg anti-PD-1 (Bio-X-Cell, BE0146, Lebanon, NH, USA) or isotype control antibody (IgG2a, Bio-X-Cell, BE0089) began and continued until euthanasia.

## Muscle lysate preparation

Frozen muscle samples were split into fragments using a mortar and pestle on dry ice and one fragment transferred to a 2 mL reinforced tube (Bertin, Montigny-le-Bretonneux, France) containing 200 μL ice cold RIPA buffer (pH 8.0, 150 mM NaCl, 50 mM Tris, 1% NP-40, 0.5% sodium deoxycholate, 0.1% SDS) with freshly added protease inhibitor cocktail (Roche, Indianapolis, USA) and 5–6 ceramic beads (2.8 mm, Bertin). Samples were then homogenized by shaking on the Precellys Evolution with Cryolys attachment (Bertin) at 7200 rpm for 2x25 sec, with 10 sec pause, at 0˚C. Lysates were spun down at 10 600 g at 4˚C for 10 minutes and the supernatant transferred to a fresh microtube. Cleared lysates were stored at -80˚C.

## Western blotting

We chose to investigate skeletal muscle mitochondrial content using Western blotting, which, although semi-quantitative, is a well-established method to assess mitochondrial content in skeletal muscle [6, 25–27]. Samples were prepared for SDS-Page by combining dithiothreitol (DTT, final conc. 0.1M), 4x LDS sample buffer (final conc. 1x), RIPA buffer with protease inhibitor cocktail and the sample (final protein conc. 2 μg/μL, determined by bicinchoninic acid (BCA) assay), before incubating for 10 minutes at 50˚C. Gels (4–12% BOLT® Bis-Tris gradient SDS gel, Invitrogen, Carslbad, CA, USA) were loaded with 25 μg protein per well. After separation, proteins were transferred to a PVDF membrane. All samples were run in duplicate on separate blots. Membranes were blocked in 5% skim milk in TBST (tris-buffered saline, TBS, with 0.1% Tween-20) at room temperature for 1h. The membrane was then incubated overnight at 4˚C with the primary antibody at the appropriate dilution (anti-COX-IV: 1:2000, Abcam, ab14744; anti-cytochrome c: 1: 1000, Thermo Fisher, 33–8500; total OXPHOS rodent WB antibody cocktail: 1:250, Abcam, ab110413; anti-GAPDH: 1:10 000, Abcam, ab181602). Next, the membrane was washed three times in TBST for 5 minutes before incubation with the secondary antibodies (anti-mouse IRDye 800CW, 1:5000, Abcam, ab216772; anti-rabbit IRDye 680RD, 1:10 000, Abcam, ab216777) in Odyssey blocking buffer and TBS (1:1) for 1h at room temperature in the dark. After a further three washing steps in TBST the membrane was dried and imaged with an Alliance Chromapure (Uvitec).

Densitometric quantification of protein bands was performed using Image J software. The signal was normalised between blots by the use of the same positive control (25 μg muscle lysate) and between samples within the same blot by the use of GAPDH as an internal loading control (refer to S1 Fig for evidence that GADH expression does not change with exercise). Specifically, the sample band intensity was first divided by the control band intensity. This value was then divided by the GAPDH band intensity for that sample to obtain the relative protein expression.

## Citrate synthase activity assay

Citrate synthase activity is frequently used as a measure of mitochondrial content in skeletal muscle [25, 28, 29]. Muscle homogenates were prepared by adding a frozen tissue fragment to 100 μL CS assay buffer (from Citrate Synthase Assay kit, Abcam, ab239712) in a microtube and homogenizing using a plastic pestle. Samples were then sonicated on ice for 10–15 seconds and cleared by spinning at 10 600 g for 10 minutes at 4˚C. Samples were stored at -80˚C. Citrate synthase activity was measured using Citrate Synthase Assay kits (Abcam, ab239712) according to the manufacturer's instructions. Briefly, 1:5 diluted samples were assayed by reaction with the diluted substrate mix (reaction mix) and measurement of the absorbance at 412 nm every 2 minutes for 30 minutes at 25˚C using a plate reader (Multiskan Sky, Thermo Fisher Scientific, Sunnyvale, CA, USA).

## Complex IV activity assay

Complex IV activity was measured using Complex IV Rodent Enzyme Activity Microplate Assay kits (Abcam, ab109911) according to the manufacturer's instructions. Muscle homogenates were also prepared according to the manufacturer's instructions. Relative activity values were obtained by normalizing to a positive control (undiluted muscle homogenate). Briefly, protein was extracted from muscle homogenates using detergent extraction and cleared by spinning at 20 776 g for 20 minutes at 4˚C. Samples were stored at -80˚C. Then, 50 μg of extracted protein was assayed by incubation for 3 hours in an anti-cytochrome c oxidase coated plate, followed by three wash steps and addition of the reaction mix (diluted cytochrome c). Absorbance was measured at 550 nm at 30˚C at a measurement interval of 2 minutes for the first 30 minutes, then at 5 minute intervals for another 90 minutes (Multiskan Sky, Thermo Fisher).

## Statistical analysis

All data were analyzed using GraphPad Prism 7 or 8. The D'Agostino-Pearson normality test was used to determine if data was from a Gaussian distribution to inform whether data should be analyzed by parametric or non-parametric test. Correlations were determined using Pearson (normally distributed) or Spearman (non-normally distributed) correlation according to the result of the normality test. Comparison between the slopes of two best-fit lines was performed using linear regression including testing for significant differences between the slopes (equivalent to analysis of covariance, ANCOVA). Comparison of the effect of one variable between two groups was performed using unpaired, two-tailed student's t test (normally distributed data) or Mann-Whitney test (non-normally distributed data) as indicated in the figure legends. Comparison of the effect of two variables and their interaction was performed using two-way ANOVA with Tukey's or Holm-Sidak's multiple comparison post-hoc tests to determine differences between individual groups if a significant main effect was found. P values less than 0.05 were considered significant. P values in text are main effects or from post-hoc analyses as indicated.

# Results

## Tumor growth, body and organ weights

We previously showed that post-implant exercise did not alter tumor-growth rate of B16-F10 or EO771 tumors [22]. Similarly, exercise did not alter tumor growth rate in mice receiving an isotype control antibody (IgG2a) or anti-PD-1 (Fig 1).

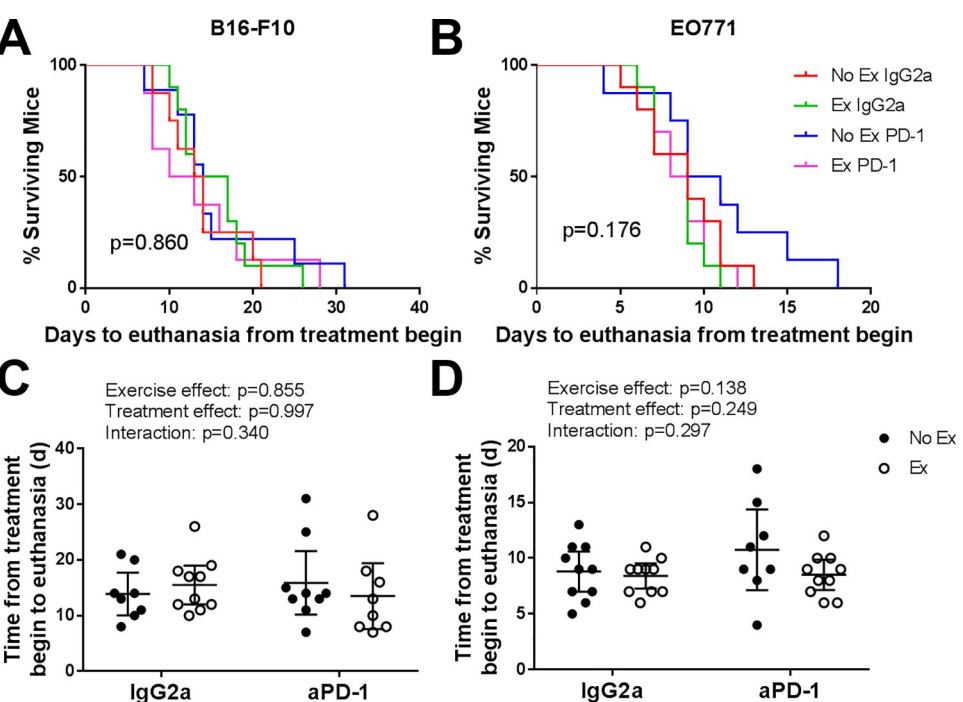

**Fig 1. Exercise and anti-PD-1 treatment do not alter growth rate of B16-F10 or EO771 tumours.** Survival curves for mice bearing B16-F10 (**a**) or EO771 (**b**) tumours (endpoint due to tumour size only). Data analysed by Log rank (Mantel-Cox) test. Time from treatment begin to euthanasia for mice bearing B16-F10 (**c**) or EO771 (**d**) tumours. B16-F10: n = 8–10 per group; EO771: n = 8–10 per group. Data are shown as individual data points and mean with 95% CI. Data analysed by two-way ANOVA.

Given that exercise can delay the onset of cachexia while not altering tumor incidence [26], we additionally investigated body weight change and heart weight. Our previous data showed that body weight change was not altered by exercise in tumor-bearing mice [22]. We showed a significant increase in heart weight in EO771-bearing mice (from 6.13 ± 0.52 g to 6.98 ± 0.52, p = 0.0008), with a similar trend for B16-F10-bearing mice (5.95 ± 0.46 g to 6.45 ± 0.88 g, p = 0.095, [22]). In order to identify whether these changes (or lack thereof) in heart weight and body weight deviate from those which are seen in healthy mice, we also investigated these parameters in tumor-free mice. Exercising tumor-free mice showed an attenuated body weight gain compared with non-exercising mice, and exercising mice tended to have increased heart weight (Table 1). Most tumor-bearing mice did not lose >5% body weight from baseline [22], indicating that they were pre-cachectic. Running distance was highly variable in both tumor-free and tumor-bearing mice (range <1 km/day– 23 km/day). Despite this, average daily running distance was similar across all groups (B16-F10-bearing mice 8.43 ± 3.17 km, EO771-bearing mice 7.81 ± 1.41 km [22], compared to tumor-free mice 7.91 ± 1.14 km, Table 1). These data indicate that the mice used in this study were largely pre-cachectic, as they did not exhibit large body weight loss or reduced running behaviour.

Anti-PD-1 resulted in less weight gain independently of exercise in mice with B16-F10 tumors (p = 0.032, Table 2) but did not affect average daily running distance, the spleen to body weight ratio, kidney to body weight ratio or the heart to body weight ratio, and the initial body weight was not different between groups (Table 2). Exercise significantly increased liver weight relative to body weight in mice with B16-F10 tumours (Table 2).

In mice with EO771 tumors, exercise significantly increased heart weight relative to body weight in an anti-PD-1 treatment independent manner (p = 0.0044, Table 3). We found no

**Table 1. Running distance, body and heart weights of tumor-free exercising vs non-exercising mice.**

| | Tumor-Free | | |
| --- | --- | --- | --- |
| | **No Ex** | **Ex** | **p** |
| Average daily running distance (km) | NA | 7.91 ± 1.14 | - |
| Initial body weight (g) | 19.1 ± 1.13 | 19.9 ± 1.69 | 0.27 |
| Final body weight (g) | 20.7 ± 1.10[a] | 20.4 ± 0.85[b] | 0.47 |
| Change in body weight (%) | 7.92 ± 3.26 | 2.89 ± 6.36 | **0.042** |
| Heart/body weight (mg/g) | 5.75 ± 0.39 | 6.17 ± 0.57 | 0.077 |

[a]Statistically significant difference between initial and final body weight in non-exercising mice (p<0.0001, paired, two-tailed student's t test).

[b]No difference between initial and final body weights in exercising mice (p = 0.30, paired, two-tailed student's t test). Values are means ± SD. p-values are for exercising (Ex) vs non-exercising (No Ex) mice for the respective group. Data were analyzed using an unpaired two-tailed student's t test. n = 9–12. Body weight has been corrected for tumor mass.

difference in average daily running distance, the spleen to body weight ratio, liver to body weight ratio, kidney to body weight ratio or body weight change with anti-PD-1 treatment, and the initial body weight was not different between groups (Table 3).

## Exercise responses of skeletal muscle mitochondria

We first investigated muscle from tumor-free mice, to establish a baseline for the effects of short-term voluntary wheel running on skeletal muscle mitochondria in a healthy model before moving on to tumor-bearing mice. In muscle from tumor-free mice, exercise significantly increased the expression of cytochrome c (p = 0.0104), complex III (p = 0.0128), complex IV (p = 0.0012) and COX-IV (p = 0.0244), while complex I, II, V, citrate synthase activity and complex IV activity were unchanged (Fig 2). This indicates that the short training period (19 days) was sufficient for some mitochondrial adaptation to training to occur, and agrees with previous data illustrating that endurance exercise, such as voluntary wheel running, induces increases in mitochondrial content [30–32].

**Table 2. Running distance, body and organ weights of B16-F10 melanoma-bearing non-exercising vs exercising mice receiving anti-PD-1 or IgG2a treatment.**

| | B16 | B16 | B16 | B16 | p-value treatment effect | p-value exercise effect | p-value interaction effect |
| --- | --- | --- | --- | --- | --- | --- | --- |
| | No Ex IgG2a | No Ex aPD-1 | Ex IgG2a | Ex aPD-1 | | | |
| Average daily running distance (km) | NA | NA | 7.30 ± 2.18 | 7.15 ± 1.14 | | 0.906[1] | |
| Initial body weight (g) | 18.1 ± 1.00 | 19.0 ± 1.52 | 18.3 ± 0.96 | 19.1 ± 1.36 | | 0.105[2] | |
| Final body weight (g) | 19.9 ± 1.13 | 19.8 ± 2.10 | 19.3 ± 0.89 | 19.4 ± 1.56 | 0.949 | 0.326 | 0.850 |
| Change in body weight (%) | 9.78 ± 9.62 | 4.00 ± 5.91 | 5.73 ± 5.59 | 1.84 ± 5.40 | **0.032*** | 0.161 | 0.665 |
| Heart/body weight (mg/g) | 6.26 ± 0.80 | 6.63 ± 0.55 | 6.51 ± 0.31 | 6.63 ± 0.55 | 0.187 | 0.486 | 0.486 |
| Spleen/body weight (mg/g) | 6.26 ± 3.92 | 4.75 ± 1.73 | 3.95 ± 0.69 | 4.40 ± 1.03 | 0.457 | 0.067 | 0.174 |
| Liver/body weight (mg/g) | 46.8 ± 2.55 | 43.6 ± 4.75 | 48.7 ± 1.61 | 48.2 ± 3.30 | 0.088 | **0.0033**** | 0.194 |
| Kidney/body weight (mg/g) | 13.6 ± 1.22 | 13.7 ± 1.00 | 13.8 ± 0.88 | 14.1 ± 0.66 | 0.524 | 0.424 | 0.659 |

Values are means ± SD. Data were analyzed using two-way ANOVA. n = 10. Body weight has been corrected for tumor mass.

[1]Test for treatment effect on running distance using unpaired, two-tailed student's t test.

[2]Test for pre-study differences between groups using Kruskal-Wallis test.

**Table 3. Running distance, body and organ weights of EO771 breast tumor-bearing non-exercising vs exercising mice receiving anti-PD-1 or IgG2a treatment.**

| | EO771 | EO771 | EO771 | EO771 | p-value treatment effect | p-value exercise effect | p-value interaction effect |
|---|---|---|---|---|---|---|---|
| | No Ex IgG2a | No Ex aPD-1 | Ex IgG2a | Ex aPD-1 | | | |
| Average daily running distance (km) | NA | NA | 7.52 ± 2.35 | 7.67 ± 1.51 | | 0.864[1] | |
| Initial body weight (g) | 17.2 ± 0.65 | 17.7 ± 1.61 | 16.9 ± 0.82 | 17.0 ± 0.78 | | 0.323[2] | |
| Final body weight (g) | 18.0 ± 0.38 | 17.9 ± 0.78 | 17.6 ± 0.71 | 18.1 ± 0.25 | 0.277 | 0.588 | 0.176 |
| Change in body weight (%) | 4.73 ± 3.79 | 1.75 ±7.06 | 4.58 ±5.86 | 6.77 ± 4.76 | 0.821 | 0.175 | 0.151 |
| Heart/body weight (mg/g) | 6.09 ± 0.55 | 6.17 ± 0.34 | 6.64 ± 0.31 | 6.44 ± 0.40 | 0.670 | **0.0044**** | 0.284 |
| Spleen/body weight (mg/g) | 3.95 ± 0.50 | 4.19 ± 0.55 | 3.78 ± 0.72 | 3.76 ± 0.92 | 0.635 | 0.195 | 0.571 |
| Liver/body weight (mg/g) | 45.5 ± 3.10 | 46.2 ± 2.60 | 47.1 ± 3.68 | 47.7 ± 5.16 | 0.614 | 0.228 | 0.979 |
| Kidney/body weight (mg/g) | 14.2 ± 0.55 | 14.3 ± 0.47 | 13.9 ± 1.10 | 13.7 ± 1.02 | 0.790 | 0.157 | 0.547 |

[1] Values are means ± SD. Data were analyzed using two-way ANOVA. n = 8–10. Body weight has been corrected for tumor mass.

[1]Test for treatment effect on running distance using unpaired, two-tailed student's t test.

[2]Test for pre-study differences between groups using one-way ANOVA

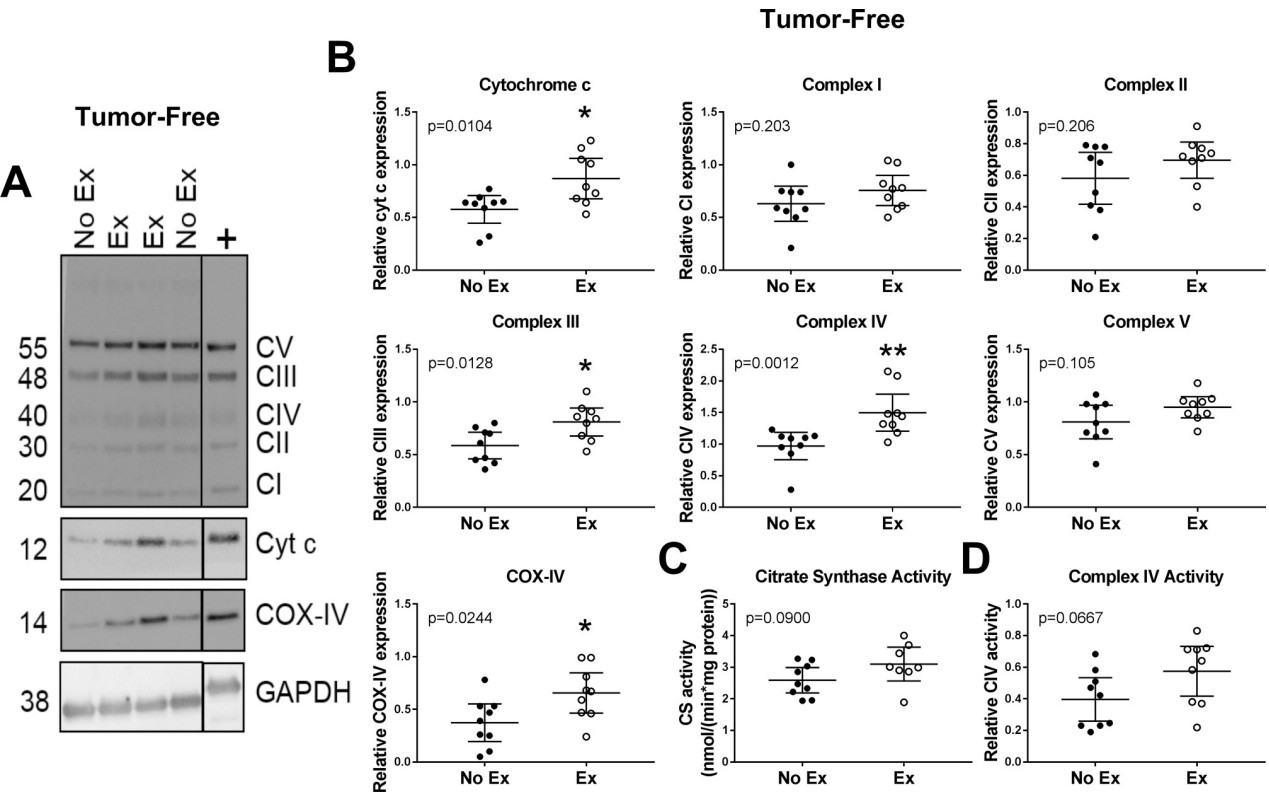

**Fig 2. Short-term exercise induces weak adaptation of skeletal muscle mitochondria in tumor-free mice. (A)** Representative Western blots of quadriceps femoris muscle homogenates from exercising or non-exercising tumor-free mice. Blots were probed for either OXPHOS complexes I-V (CI-V), cytochrome c and GAPDH, or for COX-IV and GAPDH. All samples were run in duplicate on separate gels. The positive control shown is from the same blot with the intervening lanes cropped out. GAPDH was used as loading control. **(B)** Densitometric quantification of blots in (A). Samples were normalized to a positive control run on each gel and GAPDH. Activity of citrate synthase **(C)** and complex IV **(D)** in quadriceps muscle homogenates from exercising or non-exercising tumor-free mice. Exercise effect: * indicates statistically significantly different from No Ex. Data analyzed by unpaired, two-tailed student's t test (cyt c, CI-III, CV, COX-IV, CS activity and CIV activity) or Mann-Whitney test (CIV expression). p<0.05*; p<0.01**. Data shown as individual data points and mean with 95% CI. n = 9 per group.

Local inflammation and the subsequent resolution thereof may be an essential stimulus for muscular adaptation to exercise [14]. It is currently unknown whether anti-PD-1 treatment affects exercise adaptations in skeletal muscle. Thus, we investigated how the combination of exercise and anti-PD-1 treatment affected markers of mitochondrial content in the quadriceps muscles of mice with B16-F10 or EO771 tumors.

In untreated mice with B16-F10 melanoma, skeletal muscle cytochrome c (p = 0.0026), complex II (p = 0.0264), complex III (p = 0.0122), complex IV (p = 0.0037) and complex V (p = 0.0040) levels were elevated in exercising compared with non-exercising mice, while complex I, COX-IV, citrate synthase activity and complex IV activity were unchanged (Fig 3). This suggests that the skeletal muscle mitochondrial exercise response is intact in mice bearing B16-F10 melanoma, as the response was similar to that seen in tumor-free mice (Fig 2).

As untreated mice and immune checkpoint inhibitor-treated mice were from different experiments, the data from the two cohorts were not directly compared (illustrated by a dotted line in each graph) but are presented alongside each other for visual comparison (Fig 3).

Surprisingly, we found that exercise did not increase the expression of any measured mitochondrial marker in the quadriceps femoris muscle of mice with B16-F10 melanoma receiving IgG2a treatment (Fig 3). This is in contrast to the results described above in untreated mice, where exercise induced increases in a number of markers, and suggests that the isotype control antibody used here is not inert and prevents training adaptation of skeletal muscle mitochondria in mice with B16-F10 melanoma. In mice treated with anti-PD-1, however, exercise significantly increased the expression of complex I (post-hoc p = 0.0041), complex II (post-hoc p = 0.016), complex III (post-hoc p = 0.012), complex IV (post-hoc p = 0.030), complex V (post-hoc p = 0.015) and COX-IV (post-hoc p = 0.0078; Fig 3B). Cytochrome c expression, citrate synthase activity and complex IV activity were not significantly altered by exercise in anti-PD-1-treated mice (Fig 3).

In untreated mice with EO771 breast cancer, exercise did not induce an increase in expression of cytochrome c, complex I-III, complex V or COX-IV in the quadriceps femoris muscle (Fig 4). However, complex IV expression, citrate synthase activity and complex IV activity were significantly increased by exercise (p = 0.0026, p = 0.0195 and p = 0.0083, respectively). This is in contrast to the results in tumor-free mice (Fig 2) and suggests that breast tumor-burden may alter exercise responses of skeletal muscle mitochondria.

In mice with EO771 breast cancer receiving the isotype control antibody, exercise induced robust increases in the expression of cytochrome c (post-hoc p = 0.0007), complex I (post-hoc p = 0.0040), complex II (post-hoc p = 0.0007), complex III (post-hoc p = 0.0008), complex IV (post-hoc p<0.0001), complex V (post-hoc p = 0.0035) and COX-IV (post-hoc p = 0.0029), while citrate synthase and complex IV activity were unchanged (Fig 4). In mice receiving the anti-PD-1 antibody, cytochrome c, complex I, complex III, complex IV and complex V were similarly increased by exercise (post-hoc p = 0.0011, p = 0.0059, p = 0.0091, p = 0.0010, p = 0.045, respectively; Fig 4). However, complex II and COX-IV expression were unchanged by exercise in EO771-bearing mice receiving anti-PD-1, as were citrate synthase and complex IV activity. Therefore, mice with EO771 breast cancer receiving IgG2a or anti-PD-1 appear to have a more robust exercise response compared with untreated mice, indicating that immune modulation may modify the response of skeletal muscle mitochondria to exercise training in tumor-bearing mice.

Taken together, our data provide evidence that skeletal muscle mitochondrial response to exercise training is modified by IgG2a and anti-PD-1 administration, in a tumor-type dependent manner.

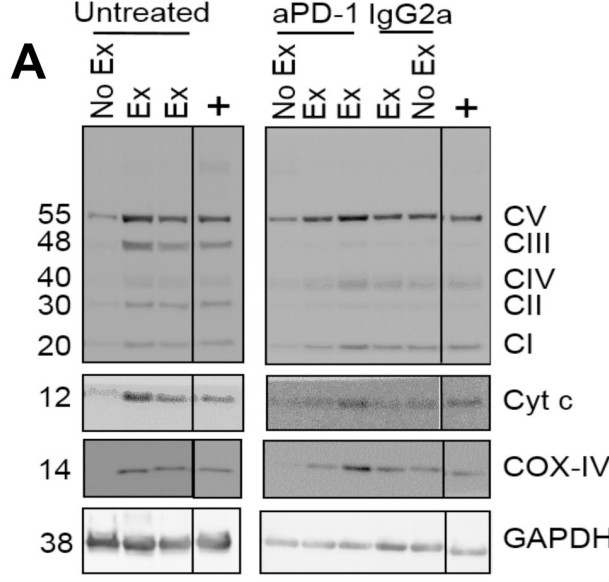

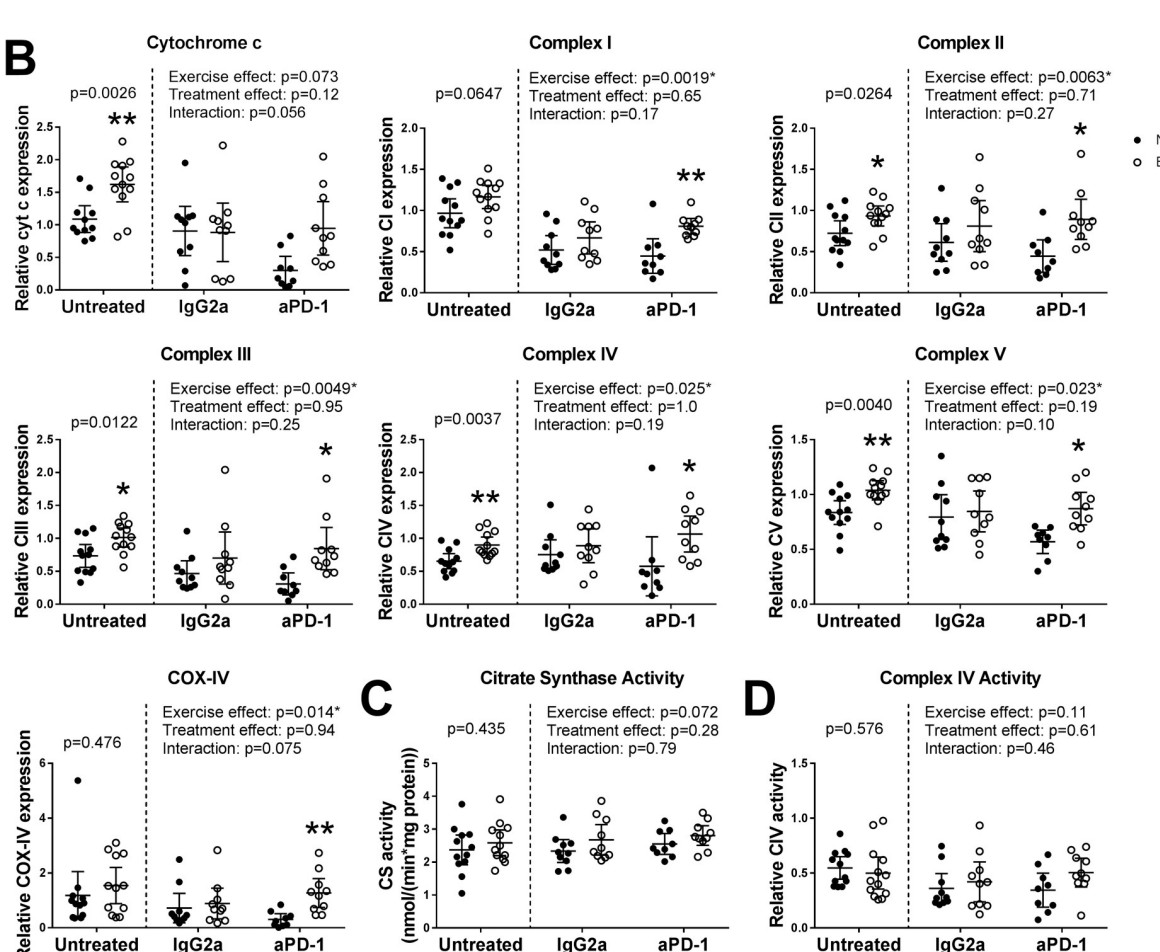

**Fig 3. Exercise increases markers of skeletal muscle mitochondria in untreated and anti-PD-1 treated mice with B16-F10 melanoma, but not mice receiving IgG2a. (A)** Representative Western blots of quadriceps femoris muscle homogenates from exercising (Ex) or non-exercising (No Ex) mice with B16-F10 melanoma receiving no treatment, an isotype control antibody (IgG2a) or anti-PD-1 (aPD-1). Blots were probed for either OXPHOS complexes I-V (CI-V), cytochrome c and GAPDH, or for COX-IV and GAPDH. All samples were run in

duplicate on separate gels. The two panels shown represent separate blots, while the dividing lines within panels indicate where lanes have been cropped out. GAPDH was used as loading control. **(B)** Relative protein expression of cytochrome c, OXPHOS complexes I-V and COX-IV in quadriceps muscle homogenates from exercising or non-exercising B16-F10 melanoma-bearing mice receiving no treatment or treated with anti-PD-1 or an isotype control antibody. Samples were normalized to a positive control (run on all blots) and GAPDH. **(C)** Activity of citrate synthase in quadriceps muscle homogenates from exercising or non-exercising mice with B16-F10 melanoma receiving no treatment or treated with anti-PD-1 or an isotype control antibody. **(D)** Activity of complex IV in quadriceps muscle homogenates from exercising or non-exercising mice with B16-F10 melanoma receiving no treatment or treated with anti-PD-1 or an isotype control antibody. *indicates statistically significantly different from No Ex (same treatment group). Data analyzed by two-way ANOVA with Sidak's multiple comparisons post-test. $p < 0.05^*$; $p < 0.01^{**}$. Data shown as individual data points and mean with 95% CI. n = 9–12 per group.

### Correlations between complex IV-associated markers

We investigated whether complex IV protein expression, complex IV activity and COX-IV (a subunit of complex IV) protein expression were associated, as this could provide insight into the assembly of the complex. Complex IV and COX-IV expression were correlated in muscle from tumor-free mice (r = 0.67, p = 0.0023; Table 4), and there tended to be a positive correlation between complex IV activity and expression (r = 0.45, p = 0.063, Table 4). In contrast, there was no correlation between these variables in tumor-bearing mice, potentially indicating a dysregulation in complex assembly in tumor-bearing mice.

In light of these results, we decided to investigate further by performing correlations between complex IV expression, complex IV activity, COX-IV and cytochrome c, the substrate for complex IV (Table 4).

In the muscle of tumor free mice, cytochrome c was strongly correlated with complex IV and COX-IV protein expression (r = 0.85, p<0.0001 and r = 0.87, p<0.0001, respectively; Table 4). In B16-F10-bearing mice, the correlation between cytochrome c and complex IV expression (although significant) was weaker (correlation: r = 0.62, p = 0.0017; comparison between slopes vs tumor-free: p = 0.028; Table 4). In mice with EO771 tumors, complex IV expression was significantly correlated with cytochrome c expression (r = 0.49, p = 0.019), and the slope was not significantly different from tumor-free mice (p = 0.29; Table 4). The correlation between cytochrome c and COX-IV was attenuated in mice with B16-F10 tumors (correlation: r = 0.54, p = 0.0078; comparison between slopes vs tumor-free: p = 0.035; Table 4) and absent in mice with EO771 tumors (comparison between slopes vs tumor-free: p = 0.043). Cytochrome c expression and complex IV activity were not correlated in muscle from either tumor-free or tumor-bearing mice (Table 4).

These data indicate that tumor burden may dysregulate complex IV assembly, potentially affecting overall function of the OXPHOS chain.We next performed the same correlations for mice receiving IgG2a or anti-PD-1. In mice with either tumor type, treated with IgG2a or anti-PD-1, complex IV expression was significantly correlated with COX-IV expression (Table 5). This is in contrast to the results described above for untreated mice, where these markers were not correlated (Table 4). Complex IV activity was not correlated with any marker (Table 5).

Similarly, cytochrome c expression was strongly correlated with both complex IV expression and COX-IV expression in mice with either tumor type, receiving IgG2a or anti-PD-1 (Table 5). In untreated mice with EO771 tumors, cytochrome c expression did not correlate with COX-IV expression (Table 4).

These results suggest that IgG2a or anti-PD-1 administration may restore the dysregulated complex IV assembly that may be occurring with tumor burden. We emphasize that this interpretation remains speculative.

## Discussion

Our study was conducted on selected markers of mitochondrial content in the skeletal muscle of mice with B16-F10 melanoma or EO771 breast cancer, with the aim of exploring the effect

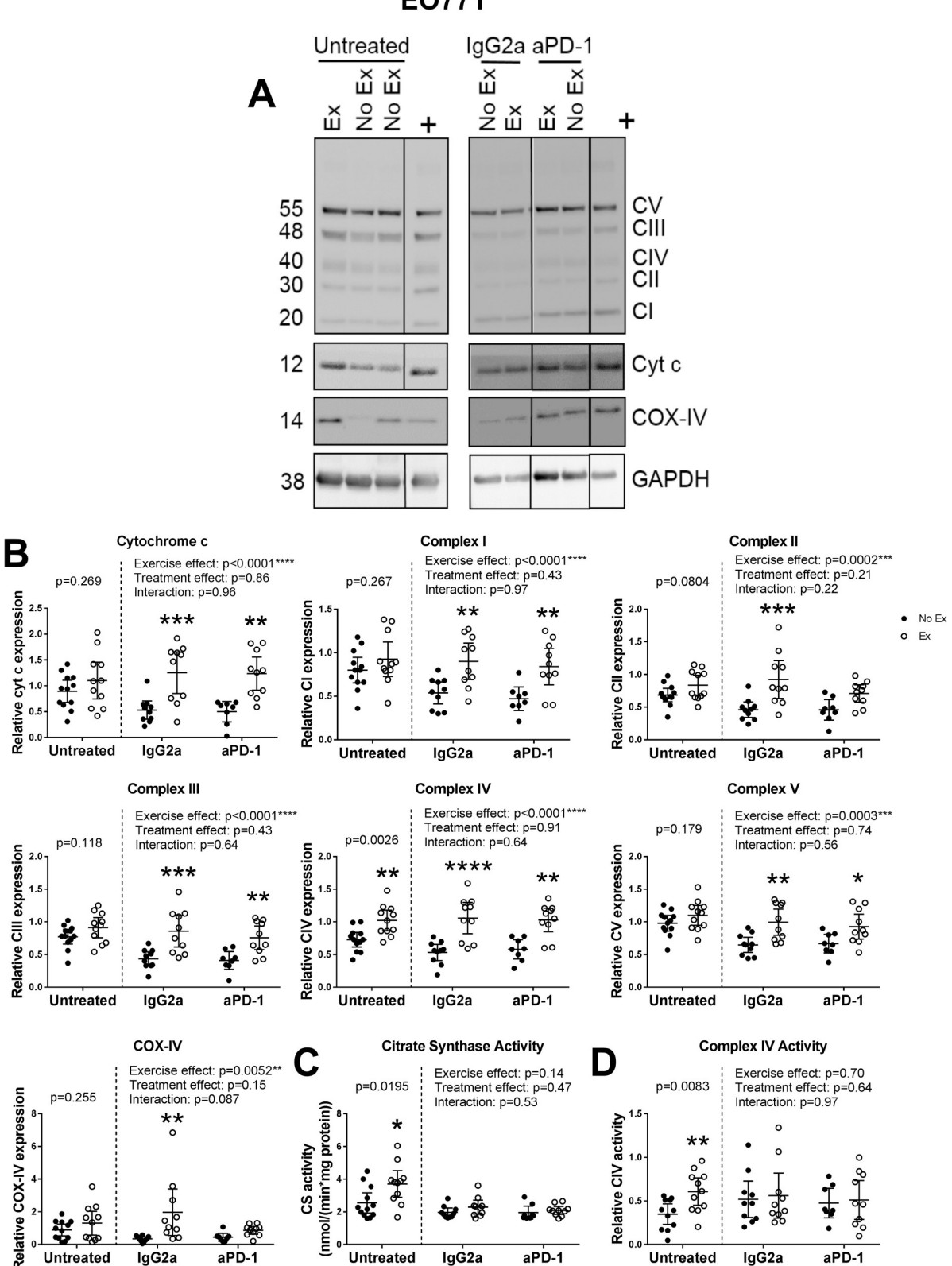

**Fig 4. Exercise induces robust increases in mitochondria marker expression in IgG2a and anti-PD-1-treated, but not untreated, mice with EO771 breast cancer. (A)** Representative Western blots of quadriceps femoris muscle homogenates from exercising (Ex) or non-

exercising (No Ex) mice with EO771 breast cancer receiving no treatment, an isotype control antibody (IgG2a) or anti-PD-1 (aPD-1). Blots were probed for either OXPHOS complexes I-V (CI-V), cytochrome c and GAPDH, or for COX-IV and GAPDH. All samples were run in duplicate on separate gels. The two panels shown represent separate blots, while the dividing lines within panels indicate where lanes have been cropped out. **(B)** Relative protein expression of cytochrome c, OXPHOS complexes I-V and COX-IV in quadriceps muscle homogenates from exercising or non-exercising EO771 breast cancer-bearing mice receiving no treatment or treated with anti-PD-1 or an isotype control antibody. Samples were normalized to a positive control (run on all blots) and GAPDH. **(C)** Activity of citrate synthase in quadriceps muscle homogenates from exercising or non-exercising mice with EO771 breast cancer receiving no treatment or treated with anti-PD-1 or an isotype control antibody. **(D)** Activity of complex IV in quadriceps muscle homogenates from exercising or non-exercising mice with EO771 breast cancer receiving no treatment or treated with anti-PD-1 or an isotype control antibody. *indicates statistically significantly different from No Ex (same treatment group). Data analyzed by two-way ANOVA with Sidak's multiple comparisons post-test. $p < 0.05^*$; $p < 0.01^{**}$, $p < 0.001^{***}$, $p < 0.0001^{****}$. Data shown as individual data points and mean with 95% CI. n = 8–12 per group.

of immunotherapy on the exercise response in muscle. These interactions are important to understand as exercise becomes increasingly promoted for people with cancer. Our data suggest that EO771 breast tumor burden impairs exercise responses of skeletal muscle mitochondria, while B16-F10 tumor burden appears to have a lesser effect. Our results suggest that both IgG2a and anti-PD-1 administration alter skeletal muscle mitochondrial response to exercise training in mice, in a tumor-type dependent manner. In mice with EO771 breast tumors in particular, both IgG2a and anti-PD-1-treated mice showed robust skeletal muscle mitochondrial exercise responses, while untreated mice did not. This raises the intriguing possibility that immune modulation may enhance exercise adaptation in tumor-bearing mice, and presents a new direction for future research.

In our untreated cohort, correlation analysis between complex IV expression, complex IV activity and COX-IV suggested that complex assembly may have been dysregulated. Similar findings were recently reported in the quadriceps muscles of pre-cachectic $Apc^{Min/+}$ mice, in which the authors found that while expression of cytochrome c, complex I and complex II increased with exercise, complex IV expression was reduced in $Apc^{Min/+}$ compared with wild-type mice, and this did not increase with exercise [26]. This suggests that complex IV may be particularly sensitive to tumor-derived factors (such as cytokines or metabolites) and may serve as an early indicator of pre-cachexia.

Inoculation with antibodies from foreign species can induce an immune response against these antibodies [33]. This may result in an altered immune response [34] and explain why mice with EO771 tumors receiving IgG2a showed an increased expression of OXPHOS proteins with exercise, while untreated mice did not. Given that the immune system plays an essential role in muscle regeneration following injury (including ultrastructural damage caused by exercise) [13], we speculate that the immune response induced by antibody inoculation synergized with exercise to enhance adaptive responses in skeletal muscle in mice with EO771

**Table 4. Correlations between complex IV markers (complex IV expression, complex IV activity, COX-IV expression) and cytochrome c expression in untreated mice.**

| | CIV activity | | | | | | COX-IV expression | | | | | | Cyt c expression | | | | | |
|---|---|---|---|---|---|---|---|---|---|---|---|---|---|---|---|---|---|---|
| | *Tumor-Free* | | *B16-F10* | | *EO771* | | *Tumor-Free* | | *B16-F10* | | *EO771* | | *Tumor-Free* | | *B16-F10* | | *EO771* | |
| | r | p | r | p | r | p | r | p | r | p | r | p | r | p | r | p | r | p |
| CIV expression | 0.45 | 0.063 | 0.015 | 0.95 | 0.084 | 0.71 | **0.67** | **0.0023** | 0.25 | 0.24 | 0.18 | 0.42 | **0.85** | **<0.0001** | **0.62** | **0.0017** | **0.49** | **0.019** |
| CIV activity | - | - | - | - | - | - | 0.32 | 0.19 | 0.25 | 0.27 | -0.16 | 0.47 | 0.38 | 0.12 | 0.00 | 0.92 | 0.27 | 0.23 |
| COX-IV expression | - | - | - | - | - | - | - | - | - | - | - | - | **0.87** | **<0.0001** | **0.54** | **0.0078** | 0.33 | 0.13 |

Data analyzed by Pearson (tumor-free, B16-F10, EO771 CIV activity vs expression, cyt c correlations) or Spearman (EO771 COX-IV vs CIV activity/expression) correlation and linear regression with comparison between slopes (ANCOVA, p values reported in main text). Tumor-Free: n = 18; B16-F10: n = 22–24; EO771: n = 22–23.

**Table 5. Correlations between complex IV markers (complex IV expression, complex IV activity, COX-IV expression) and cytochrome c expression in tumor-bearing mice receiving either iso-type control (IgG2a) or anti-PD-1 (aPD-1) treatment.**

| | CIV activity | | | | | | | | COX-IV expression | | | | | | | | Cyt c expression | | | | | | | |
| | B16-F10 | | | | EO771 | | | | B16-F10 | | | | EO771 | | | | B16-F10 | | | | EO771 | | | |
| | IgG2a[a] | | aPD-1[a] | | IgG2a[b] | | aPD-1[a] | | IgG2a[b] | | aPD-1[b] | | IgG2a[b] | | aPD-1[a] | | IgG2a[a] | | aPD-1[b] | | IgG2a[a] | | aPD-1[a] | |
| | r | p | r | p | r | p | r | p | r | p | r | p | r | p | r | p | r | p | r | p | r | p | r | p |
|---|---|---|---|---|---|---|---|---|---|---|---|---|---|---|---|---|---|---|---|---|---|---|---|---|
| **CIV expression** | -0.36 | 0.12 | -0.058 | 0.81 | -0.071 | 0.76 | -0.27 | 0.28 | 0.78 | <0.0001 | 0.86 | <0.0001 | 0.82 | <0.0001 | 0.76 | 0.0002 | 0.76 | 0.0001 | 0.88 | <0.0001 | 0.90 | <0.0001 | 0.79 | 0.0001 |
| **CIV activity** | - | - | - | - | - | - | - | - | -0.35 | 0.13 | 0.23 | 0.34 | 0.085 | 0.72 | -0.26 | 0.30 | -0.10 | 0.68 | 0.13 | 0.59 | -0.26 | 0.27 | -0.077 | 0.76 |
| **COX-IV expression** | - | - | - | - | - | - | - | - | - | - | - | - | - | - | - | - | 0.53 | 0.015 | 0.84 | <0.0001 | 0.68 | 0.0009 | 0.61 | 0.0067 |

Data analyzed by Pearson[a] or Spearman[b] correlation. B16-F10 IgG2a: n = 20; B16-F10 aPD-1: n = 19; EO771 IgG2a: n = 20; EO771 aPD-1: n = 18.

tumors, independently of the specific action of anti-PD-1. Therefore, IgG2a may be acting as an immune-modulatory factor in this study, rather than as an inert control.

Tumor-free mice showed a relatively weak mitochondrial exercise response in this study, with only 4/9 markers being significantly elevated compared with non-exercising mice. This is not entirely unexpected, as the training period was only 19 days for this group and rodent exercise studies will typically use an intervention period of at least 4 weeks to induce a robust adaptive effect (our study was unable to run for longer due to tumors reaching maximum ethical size) [30, 35]. Similarly, untreated mice with B16-F10 or EO771 tumors showed increases in some markers with exercise but not others. In contrast, exercise induced a robust increase in the expression of almost all measured markers in anti-PD-1 treated mice with B16-F10 or EO771 tumors, suggesting that immune modulation may improve exercise responses of skeletal muscle mitochondria in tumor-bearing mice. This hypothesis is based on indirect comparisons between untreated and anti-PD-1-treated mice rather than comparison between IgG2a-treated and anti-PD-1 treated mice, as IgG2a appeared to have effects of its own. Unfortunately, we were unable to directly compare untreated with treated mice, as these experiments were from different studies and not conducted alongside one another. Therefore, our results should be viewed as hypothesis-generating: that anti-PD-1 enhances exercise responses in tumor-bearing mice. The implication is that if true, this provides a novel avenue of research for the prevention of cancer-associated muscle wasting, as the combination of exercise and anti-PD-1 may be more effective than exercise alone at inducing muscle adaptation.

Systemic, chronic inflammation occurs with tumor burden [36], and systemic immune dysfunction characterized by impaired effector response has been suggested as an early indicator for cancer cachexia (i.e. before clinical signs such as weight loss become apparent) [37]. This is reminiscent of T cell exhaustion, which commonly occurs in the tumor microenvironment due to persistent antigen exposure/inflammation and is characterized by high expression of inhibitory receptors (such as PD-1) and impaired effector functions [38]. We therefore speculate that tumor-associated inflammation may be inducing an exhausted phenotype in intramuscular immune cells, which is alleviated by anti-PD-1, allowing the cells to respond to a stimulus (exercise) and induce muscle adaptation more effectively.

We suggest that the differing effects of IgG2a administration on mice with B16-F10 and EO771 tumors are due to differences in how the immune system responds to the isotype control antibody in mice with either of the two tumor types. In support of this, IgG2a administration in the mice used here led to the formation of anti-rat antibodies in mice with B16-F10 but not EO771 tumors [34], but nevertheless significantly reduced the proportion of CD8$^+$ T cells within the tumor in both tumor types (manuscript under review). This is an indication that the type of immune response being induced by IgG2a administration is different in mice with EO771 compared with B16-F10 tumors and provides an explanation as to why the effect seen on the skeletal muscle mitochondrial response to exercise was also different. In order to investigate how the immune response differs in mice with the two tumor types, an initial step could be to profile circulating immune cell phenotypes in mice with either tumor type treated with IgG2a, anti-PD-1 or receiving no treatment.

Interestingly, the response of (untreated) tumor-bearing mice to exercise varied from that seen in tumor-free mice, with different markers being increased with exercise in each group. Importantly, the regulation of OXPHOS complex assembly is highly complex, and can adapt according to metabolic demand [39]. This can occur in both healthy and disease states–indeed, our data suggest that complex IV assembly may be dysregulated by tumor burden. Therefore, it is possible that the variation between groups that we see in the skeletal muscle mitochondrial response to exercise in this study is caused by differing systemic effects caused by the different tumor types, and may reflect changes in the regulation of the OXPHOS complexes to adjust

(or attempt to adjust) for changes in metabolic demand. These changes may affect mitochondrial and muscle function, and should be followed up on in the future.

## Limitations

This study was an exploratory investigation into differences in mitochondrial adaptation to exercise in tumor-bearing mice receiving immunotherapy or no treatment, and arose as an off-shoot of our main study investigating the effects of exercise and immunotherapy on the tumor microenvironment (manuscript in preparation). As such, we acknowledge the lack of a number of measures that would be required to properly investigate how immunotherapy affects muscle responses to exercise and to test the hypothesis we suggest here (that reactivation of the immune response enhances muscular exercise responses in tumor-bearing mice). These include measures of *in vivo* muscle function (strength and fatigue), exercise capacity testing, whole-body metabolism, additional indices of cachexia (inflammatory markers, food intake, muscle mass), additional measures of mitochondrial function (e.g. respirometry, markers of dynamics and biogenesis), measures of apoptosis (e.g. investigation of the subcellular localization of cytochrome c) and intramuscular immune cell populations, as well as direct comparison of tumor-free, untreated and anti-PD-1-treated mice. We would like to emphasize that this study was conducted as a hypothesis-generating foray into an as yet largely unexplored field, and we hope that the results (although preliminary) will spark future research into this area. We also acknowledge that, due to the exploratory nature of our study, there may be other interpretations of our data than those which we have presented here.

## Conclusion

Our data provide preliminary evidence that systemic immune stimulation by IgG2a or anti-PD-1 may enhance the adaptation of skeletal muscle mitochondria to exercise training. Although these results should be viewed as hypothesis-generating and followed up with more comprehensive analyses of training adaptations, they have raised the exciting possibility that exercise may be particularly effective in patients treated with immunotherapy, and thus may be able to prevent cancer-associated muscle wasting in this population.

## Supporting information

**S1 Fig. Muscular GAPDH protein expression is not altered by exercise.** Representative whole blots probed for GAPDH of muscle lysates from tumor-free mice (a) or mice with B16-F10 (b) or EO771 tumors (c), and the densitometric quantification thereof (d-f). n = 8–12 per group. Data are shown as individual data points and mean with 95% CI. Data analysed by unpaired, two-tailed student's t test (tumor-free and B16-F10) or Mann-Whitney test (EO771).
(PDF)

**S1 Raw images. Whole blot images used in Figs 2–4.**
(PDF)

## Acknowledgments

We gratefully acknowledge Mr Andrew Dachs (Decision Consulting Ltd, NZ) for developing the mouse wheels and A/Prof Andreas Moeller (QIMR Berghofer, Australia) for gifting the EO771 cells.

## Author Contributions

**Conceptualization:** Linda A. Buss, Gabi U. Dachs.

**Formal analysis:** Linda A. Buss, Troy L. Merry.

**Funding acquisition:** Linda A. Buss, Bridget A. Robinson, Margaret J. Currie, Gabi U. Dachs.

**Investigation:** Linda A. Buss.

**Methodology:** Linda A. Buss.

**Supervision:** Barry Hock, Abel D. Ang, Bridget A. Robinson, Margaret J. Currie, Gabi U. Dachs.

**Visualization:** Linda A. Buss.

**Writing – original draft:** Linda A. Buss.

**Writing – review & editing:** Linda A. Buss, Barry Hock, Troy L. Merry, Abel D. Ang, Bridget A. Robinson, Margaret J. Currie, Gabi U. Dachs.

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
