## [Decision Letter · Decision Letter 0]

22 Apr 2021

PONE-D-21-06002

Effect of Immune Modulation on the Skeletal Muscle Mitochondrial Exercise Response: An Exploratory Study in Mice with Cancer

PLOS ONE

Dear Dr. Buss,

Thank you for submitting your manuscript to PLOS ONE. After careful consideration, we feel that it has merit but does not fully meet PLOS ONE’s publication criteria as it currently stands. Therefore, we invite you to submit a revised version of the manuscript that addresses the points raised during the review process.

We look forward to receiving your revised manuscript.

Kind regards,

Atsushi Asakura, Ph.D

Academic Editor

PLOS ONE

Journal Requirements:

2.Thank you for stating the following in the Financial Disclosure section:

"We appreciate funding from the Mackenzie Charitable Foundation (MJC, GUD, BAR), the University of Otago (doctoral scholarship for LAB) and Saunders Robinson Brown (McGee Fellowship for LAB). The funders had no role in study design, data collection and analysis, decision to publish, or preparation of the manuscript."

We note that you received funding from a commercial source:  Saunders Robinson Brown.

Reviewers' comments:

Reviewer's Responses to Questions

**Comments to the Author**

1. Is the manuscript technically sound, and do the data support the conclusions?

Reviewer #1: Yes

Reviewer #2: No

2. Has the statistical analysis been performed appropriately and rigorously? 

Reviewer #1: Yes

Reviewer #2: I Don't Know

3. Have the authors made all data underlying the findings in their manuscript fully available?

Reviewer #1: Yes

Reviewer #2: Yes

4. Is the manuscript presented in an intelligible fashion and written in standard English?

Reviewer #1: Yes

Reviewer #2: Yes

5. Review Comments to the Author

Reviewer #1: I enjoyed reading your manuscript. I believe it has been designed well and by itself raised more questions.

One question that I have: You showed that tumor burden may dysregulate complex IV assembly and affecting overall function of the OXPHOS chain. Does anti-PD1 or any antibody (IgG2) have any effect (positive or negative) on this?

Reviewer #2: The authors conducted an exploratory study to investigate how tumor burden and cancer treatment with anti-PD-1 or with isotype control antibody (IgG2a), modify the skeletal muscle mitochondrial response to exercise (19 days of free-wheel running) in mice with transplantable tumors (B16-F10 melanoma and EO771 breast cancer). Mice remained sedentary or were provided with running wheels for ~19 days.

The manuscript shows that exercise and anti-PD-1 did not alter the growth rate of either tumor type, either alone or in combination therapy. However, in mice with B16-F10 tumors

exercise and anti-PD-1 therapy showed an increase in mitochondrial content markers in skeletal muscle.

Their results suggests also that the combination therapy is crucial, because mice with B16-F10 treated with control antibody and exercise did not exhibit increase in marker of mitochondrial content in skeletal muscle.

In untreated mice with EO771 tumors, a mild increase in some mitochondrial markers were observed after exercise but showed more increased markers when exercise were accompanied with anti-PD1 or IgG2a administration. and anti-PD-1-treated groups both showed robust increases in most measured markers following exercise.

In mice with EO771 tumors, both IgG2a and anti-PD-1-treated mice show robust skeletal muscle mitochondrial exercise responses, while untreated mice do not. The author postulate that immune modulation may enhance skeletal muscle mitochondrial response to exercise in tumor-bearing mice.

Remarks:

Line 56:

“prior to any noticeable muscle wasting in both mice [2] and humans [3], suggesting that mitochondrial dysfunction plays a role in the pathogenesis of cancer cachexia”

There is evidence in ref 2, for a dysfunction of mitochondrial function before a measurable mass loss. This does not mean that there is a suggestion of a possibly causality between mitochondrial disfunction and muscle loss in cachexia. Others researcher have made an extensive review for the possible pathways that induce mitochondrial dysfunction in cancer cachexia (BBA - Reviews on Cancer 1870 (2018) 137–150). Their revision of multiple available data indicates that cancer is a process able to induce cachexia and mitochondrial dysfunction. Even if this state may constitute a negative loop for muscle loos, I suggest the author of the present manuscript to be careful with sentences that implies causality.

Line 91

Due to the role of the immune system in muscle regeneration and adaptation to exercise, it is possible that immunotherapies such as anti-PD-1 will affect muscle responsiveness to exercise training.

A couple of references are needs here. Also, it has to be clear if the cellular mechanism of this role of immune system to adaptation of skeletal muscle to exercise have been reported or if the evidence is only circumstantial.

Results:

In table 1, changes in weight of mice are shown. No significant changes were reported between mice before and post exercise for initial weight and final weight. However, authors reported a significant change in % of body weight change. This result is not coherent with the first data. If change of weight is a quotient between initial and final weight, how could appear a significance? An error in statistical analysis appears most likely. If not, please explain.

Is important to treat exercise as a multifactorial stimulus, in particular in reference to endurance or strengthen exercise type. In the works used as references, clearly used an endurance type exercise, as the pathways activated correspond to those obtained with tonic electrostimulation or endurance training, exercise that induce plasticity changes from fast type to slow type muscle fibers, which have a higher content in mitochondria a higher level of mitochondrial enzymes related to oxidative metabolism. So, the first experiment that the authors should be do, is a control of the modification of muscle type and/or mitochondria content and function, obtained with the exercise in wheel. Otherwise, we can’t know if the exercise choose is, in control animals, inducing the expected results.

My main critic to this work is the conceptual design referring to exercise. If the author were searching for an exercise type that increases mitochondrial contain, they should in the first place demonstrate that the exercise chosen induced the change in mitochondrial content and function searched. This is not done. The markers for mitochondria are poor and the techniques used are weak also. I miss muscle slides to evaluate the histology of the muscle, the infiltration content, fiber integrity, fiber cross sectional area and also other markers as SDH, LDH, and general master genes as PGC1-alpha.

So, the initial and crucial step for the study is weak. The results obtained from that point pay the price of this initial design. Results on mitochondria function or content are weak and is difficult to conclude if the treatment with anti-PD1 has a crosstalk with exercise effect or not. Moreover, the title Effect of Immune Modulation on the Skeletal Muscle Mitochondrial Exercise Response, appears disconnected with the study, and too much ambitious.

In conclusion, is an important study, that lacks a proper design and execution to answer the question proposed.

Additionally, cachexia seems to lose importance along the manuscript. Indeed, Figure 1 indicates that exercise and anti-PD-1, do not alter growth rate of B16-F10 or EO771 tumors. So, we are somewhat lost in the parameter important to be evaluated.

The authors recognize the limitation of the present study, mentioning some of the points expressed above. In my opinion, some of the experiments suggested in their Discussion must be part of the present manuscript.

One of the conclusions of this study is that “in mice with B16-F10 tumors, IgG2a administration prevents exercise adaptation of skeletal muscle mitochondria, but adaptation remains intact in mice receiving anti-PD-1”. This is clearly one interpretation of the data but is not the only one. The connection of this conclusion with this other: “In mice with EO771 tumors, both IgG2a and anti-PD-1-treated mice show robust skeletal muscle mitochondrial exercise responses, while untreated mice do not” is not present in the manuscript. There is a difference between different tumor types? What is the possible hypothesis for that? What are the control experiments that must be done for probe these hypothesis?

6. PLOS authors have the option to publish the peer review history of their article (what does this mean?). If published, this will include your full peer review and any attached files.

Reviewer #1: No

Reviewer #2: **Yes: **Mariana Casas

---

## [Author Response · Author response to Decision Letter 0]

4 May 2021

Please refer to the accompanying cover letter for our detailed responses to the reviewer and editor comments.

---

## [Decision Letter · Decision Letter 1]

8 Jul 2021

PONE-D-21-06002R1

Effect of immune modulation on the skeletal muscle mitochondrial exercise response: An exploratory study in mice with cancer

PLOS ONE

Dear Dr. Buss,

Thank you for submitting your manuscript to PLOS ONE. After careful consideration, we feel that it has merit but does not fully meet PLOS ONE’s publication criteria as it currently stands. Therefore, we invite you to submit a revised version of the manuscript that addresses the points raised during the review process.

We look forward to receiving your revised manuscript.

Kind regards,

Atsushi Asakura, Ph.D

Academic Editor

PLOS ONE

Reviewers' comments:

Reviewer's Responses to Questions

**Comments to the Author**

1. If the authors have adequately addressed your comments raised in a previous round of review and you feel that this manuscript is now acceptable for publication, you may indicate that here to bypass the “Comments to the Author” section, enter your conflict of interest statement in the “Confidential to Editor” section, and submit your "Accept" recommendation.

Reviewer #2: All comments have been addressed

2. Is the manuscript technically sound, and do the data support the conclusions?

Reviewer #2: Partly

3. Has the statistical analysis been performed appropriately and rigorously? 

Reviewer #2: Yes

4. Have the authors made all data underlying the findings in their manuscript fully available?

Reviewer #2: Yes

5. Is the manuscript presented in an intelligible fashion and written in standard English?

Reviewer #2: Yes

6. Review Comments to the Author

Reviewer #2: Reviewer #2:

Line 56:

“prior to any noticeable muscle wasting in both mice [2] and humans [3], suggesting that

mitochondrial dysfunction plays a role in the pathogenesis of cancer cachexia”

There is evidence in ref 2, for a dysfunction of mitochondrial function before a measurable

mass loss. This does not mean that there is a suggestion of a possibly causality between

mitochondrial disfunction and muscle loss in cachexia. Others researcher have made an

extensive review for the possible pathways that induce mitochondrial dysfunction in cancer

cachexia (BBA - Reviews on Cancer 1870 (2018) 137–150). Their revision of multiple

available data indicates that cancer is a process able to induce cachexia and mitochondrial

dysfunction. Even if this state may constitute a negative loop for muscle loos, I suggest the

author of the present manuscript to be careful with sentences that implies causality.

Our aim with this sentence was not to imply that mitochondrial dysfunction causes

muscle mass loss, but rather to suggest that this is one of the earliest processes being

dysregulated by tumour burden and therefore may be an early indicator, as it is

present before the usual clinical indicators such as muscle mass loss.

In response, we have amended the sentence to read “…suggesting that

mitochondrial dysfunction may play a role in the pathogenesis of cancer cachexia

and serve as an early indicator of its onset” (lines 65 and 66), which we hope is

clearer.

R: OK

Line 91

Due to the role of the immune system in muscle regeneration and adaptation to exercise, it is possible that immunotherapies such as anti-PD-1 will affect muscle responsiveness to

exercise training.

A couple of references are needs here.

We have inserted references as requested.

R: Thanks.

Also, it has to be clear if the cellular mechanism of this role of immune system to adaptation

of skeletal muscle to exercise have been reported or if the evidence is only circumstantial.

This has been discussed and referenced in the preceding paragraph (lines 102-112).

We have amended the text to clarify this issue (lines 119-120).

R:Thanks

Results:

In table 1, changes in weight of mice are shown. No significant changes were reported

between mice before and post exercise for initial weight and final weight. However, authors

reported a significant change in % of body weight change. This result is not coherent with the first data. If change of weight is a quotient between initial and final weight, how could appear a significance? An error in statistical analysis appears most likely. If not, please explain.

The difference in statistical significance can be explained by the fact that the change

in body weight is a measure for the difference between initial and final body weight

within a group, which has then been compared between exercising and non-exercising

groups, while the comparison between initial and final weights across exercising and

non-exercising mice do not incorporate this difference. To further illustrate this, we

have performed a paired t test between initial and final body weight for non exercising (p<0.0001) and exercising (p=0.30) mice (Table 1 has been updated).

As can be seen, the test shows that the increase in body weight in non-exercising mice

is statistically significant, while that in exercising mice is not. This is what is being

shown by the statistically significant difference in % change in body weight between

non-exercising and exercising mice.

R:Thanks for the explanation. However, the data of this table serve to compare with previous work, and I found that rationale of this comparison is not clear in the text.

Is important to treat exercise as a multifactorial stimulus, in particular in reference to

endurance or strengthen exercise type. In the works used as references, clearly used an

endurance type exercise, as the pathways activated correspond to those obtained with tonic

electrostimulation or endurance training, exercise that induce plasticity changes from fast

type to slow type muscle fibers, which have a higher content in mitochondria a higher level

of mitochondrial enzymes related to oxidative metabolism. So, the first experiment that the

authors should be do, is a control of the modification of muscle type and/or mitochondria

content and function, obtained with the exercise in wheel. Otherwise, we can’t know if the

exercise choose is, in control animals, inducing the expected results.

We had performed this experiment, as suggested. This initial experiment is shown in

Figure 2, where we have investigated the effect of exercise on the expression of

mitochondrial markers and activity of mitochondrial enzymes in tumour-free control

mice. This was then used as a benchmark to interpret our results in tumour-bearing

mice (see for example lines 341-343, “This suggests that the skeletal muscle

mitochondrial exercise response is intact in mice bearing B16-F10 melanoma, as the

response was similar to that seen in tumor-free mice”). As this was an exploratory,

biochemical analysis, the characterisation of muscle fibre type was beyond the scope

of this investigation and we no longer have sufficient samples available for

retrospective analysis. However, we agree that this would be a valuable analysis to

add to any future investigation.

R: Thanks for your clarification on this point. Indeed, in Fig2 appears measurement by WB for 9 markers of mitochondrial content, and 4 of them present a difference with non-trained mice. But the response of tumor bearing mice was different, because B16-F10 bearing mice shows 5 increased mitochondrial markers content, with only sharing increased amounts of cytochrome c, compared to non-tumor bearing mice. All of markers of B16-F10 already increased were increased by treatment with aPD-1. If only cytochrome c is shared, effect of possible apoptosis appearance should be commented. Moreover, the difference in increased markers with exercise between non-tumor and tumor bearing mice must be explained or commented.

My main critic to this work is the conceptual design referring to exercise. If the author were

searching for an exercise type that increases mitochondrial contain, they should in the first

place demonstrate that the exercise chosen induced the change in mitochondrial content and

function searched. This is not done.

Please see our response to your previous comment, illustrating that we have, in fact,

shown a change in mitochondrial content with our chosen exercise type in control

mice (Figure 2). Additionally, it is extensively documented in the literature that

endurance exercise, such as voluntary wheel running, induces increases in

mitochondrial content. We have added and referenced lines 314-315 to clarify this

in the manuscript. Therefore, we are confident that our chosen exercise type induced

the required changes. We note that our training duration was relatively short, but as

we show in Figure 2, it was nevertheless sufficient to induce some increases in

mitochondrial content markers.

R: As you told, some of them, not in a strongly manner and more important, not the same set for the different groups used.

The markers for mitochondria are poor and the techniques used are weak also. I miss muscle slides to evaluate the histology of the muscle, the infiltration content, fiber integrity, fiber cross sectional area and also other markers as SDH, LDH, and general master genes as

PGC1-alpha.

The chosen markers relate to an essential mitochondrial function, namely oxidative

phosphorylation, and so reflect changes in mitochondrial health. The techniques

(Western blot and enzyme assays) are well-established methods commonly used in

the field. We have clarified this in the manuscript, lines 177-179 and 205-206.

We accept that Western blotting is semi-quantitative, but we have used the same

positive control sample on all blots, as well as a loading control, to make the results as

accurate and comparable to each other as possible.

Now that we have shown, for the first time, that mitochondrial function may be

altered by both tumour burden and immune modulation, follow-up studies along the

lines suggested by the reviewer are appropriate.

R: I am not questioning the validity of WB, but missing the measurements indicated.

So, the initial and crucial step for the study is weak. The results obtained from that point pay the price of this initial design. Results on mitochondria function or content are weak and is difficult to conclude if the treatment with anti-PD1 has a crosstalk with exercise effect or not. Indeed, it appears that treatment with anti-PD1

Moreover, the title Effect of Immune Modulation on the Skeletal Muscle Mitochondrial

Exercise Response, appears disconnected with the study, and too much ambitious.

Our first step was to analyse and confirm adaptation of mitochondrial function

in response to the chosen exercise regime in tumour-free mice (Figure 2, added

lines 307-309). We used that basic information to investigate the impact of tumour

burden on response to exercise and to design our exploratory study on how immune

modulators influenced mitochondrial function in this model system. This has never

been attempted before.

As an exploratory study, we agree that our results are not conclusive, but rather

hypothesis-generating. This has been clearly stated throughout the manuscript (title,

lines 31, 107-109, 563-567, 575-578, 580-582). If our initial findings are confirmed it

would have implications for understanding the development of cachexia and the role

of exercise in preventing cachexia. This has never been investigated in either animals

or humans, and our results suggest that this is a worthy avenue of future research.

Similarly, we are unsure as to why our title appears disconnected with the study and

too ambitious. We have not made any claims in the title as to how immunotherapy

might affect the exercise response, and we have included the phrase “an exploratory

study in mice with cancer” to emphasise that our results are preliminary.

R: If PLOS ONE are willing to publish “exploratory studies” is a matter od the Editor and not of the reviewer.

In conclusion, is an important study, that lacks a proper design and execution to answer the

question proposed.

We agree that our exploratory study does not conclusively answer the question of

whether immunotherapy and cancer alter the skeletal muscle response to exercise, and

that it should be followed up with more comprehensive investigations. This

limitation has been clearly stated in lines 563-578. However, we believe that our

work has merit as hypothesis-generating data for other researchers to build on, as it is

the first to investigate whether immunotherapy alters skeletal muscle mitochondria.

Additionally, cachexia seems to lose importance along the manuscript. Indeed, Figure 1

indicates that exercise and anti-PD-1, do not alter growth rate of B16-F10 or EO771 tumors. So, we are somewhat lost in the parameter important to be evaluated.

An accelerated tumour growth rate is not necessarily concurrent with the development

of cachexia. For example, it has been shown that exercise can delay the onset of

cachexia, while not altering tumour incidence (Vanderveen 2020). Therefore, it is

valid to investigate changes in cachectic indicators even in the absence of a change in

tumour growth rate. We have added lines 259-260 to clarify this. In addition, our

mice were largely pre-cachectic (line 272), which is why cachexia is not a large focus

of the manuscript.

R: OK

The authors recognize the limitation of the present study, mentioning some of the points

expressed above. In my opinion, some of the experiments suggested in their Discussion must be part of the present manuscript.

It appears to us that the main limitation for this reviewer was a weak initial step; we

hope we have been able to convince the reviewer that this initial step has in fact been

carried out in the form of analysis of mitochondrial function in tumour-free, untreated

mice (Figure 2). The suggested experiments are clearly beyond the scope of a major

revision of this manuscript as they would require a significant additional time and

funding commitment.

R: Thanks for your response, but I am not convinced with your data, for the reasons already mentioned in terms of techniques, markers (citrate synthase activity has been used as gold standard in free wheel training), interpretation of data (different markers are increased in non-tumor and tumor bearing mice, even between different tumors).

One of the conclusions of this study is that “in mice with B16-F10 tumors, IgG2a

administration prevents exercise adaptation of skeletal muscle mitochondria, but adaptation

remains intact in mice receiving anti-PD-1”. This is clearly one interpretation of the data but is not the only one. The connection of this conclusion with this other: “In mice with EO771 tumors, both IgG2a and anti-PD-1-treated mice show robust skeletal muscle mitochondrial exercise responses, while untreated mice do not” is not present in the manuscript. There is a difference between different tumor types? What is the possible hypothesis for that? What are the control experiments that must be done for probe these hypothesis?

This is a very interesting point, and we have added a paragraph discussing one

possible hypothesis as to why IgG2a has different effects on mice with B16-F10

compared with EO771 tumors (lines 550-562).

R: Thanks for include it.

7. PLOS authors have the option to publish the peer review history of their article (what does this mean?). If published, this will include your full peer review and any attached files.

Reviewer #2: No

---

## [Editor Report · Decision Letter 2]

7 Oct 2021

Effect of immune modulation on the skeletal muscle mitochondrial exercise response: An exploratory study in mice with cancer

PONE-D-21-06002R2

Dear Dr. Buss,

We’re pleased to inform you that your manuscript has been judged scientifically suitable for publication and will be formally accepted for publication once it meets all outstanding technical requirements.

Kind regards,

Atsushi Asakura, Ph.D

Academic Editor

PLOS ONE
---

## [Editor Report · Acceptance letter]

11 Oct 2021

PONE-D-21-06002R2 

Effect of immune modulation on the skeletal muscle mitochondrial exercise response: An exploratory study in mice with cancer 

Dear Dr. Buss:

I'm pleased to inform you that your manuscript has been deemed suitable for publication in PLOS ONE. Congratulations! Your manuscript is now with our production department. 

Kind regards, 

on behalf of

Dr. Atsushi Asakura 

Academic Editor

PLOS ONE